# Early IgE Production Is Linked with Extrafollicular B- and T-Cell Activation in Low-Dose Allergy Model

**DOI:** 10.3390/vaccines10060969

**Published:** 2022-06-17

**Authors:** Dmitrii Borisovich Chudakov, Olga Dmitrievna Kotsareva, Maryia Vladimirovna Konovalova, Daria Sergeevna Tsaregorodtseva, Marina Alexandrovna Shevchenko, Anton Andreevich Sergeev, Gulnar Vaisovna Fattakhova

**Affiliations:** 1Laboratory of Cell Interactions, Shemyakin-Ovchinnikov Institute of Bioorganic Chemistry, 16/10 Miklukho-Maklaya St., 117997 Moscow, Russia; olga.kotsareva@gmail.com (O.D.K.); mariya.v.konovalova@gmail.com (M.V.K.); mshevch@gmail.com (M.A.S.); cheburatorka@gmail.com (A.A.S.); gfattakhova@yahoo.com (G.V.F.); 2Faculty of Medical Biology, Sechenov First Moscow State Medical University, 2 Bolshaya Pirogovskaya St., 1194535 Moscow, Russia; tsaregorodtseva.daria@yandex.ru

**Keywords:** IgE, extrafollicular response, subcutaneous fat, plasmablasts, extrafollicular T helpers

## Abstract

Despite its paramount importance, the predominant association of early IgE production with harmless antigens, via germinal-center B- and T-cell subpopulations or extrafollicular activation, remains unresolved. The aim of this work was to clarify whether the reinforced IgE production following the subcutaneous immunization of BALB/c mice with low antigen doses in withers adipose tissue might be linked with intensified extrafollicular or germinal-center responses. The mice were immunized three times a week for 4 weeks in the withers region, which is enriched in subcutaneous fat and tissue-associated B cells, with high and low OVA doses and via the intraperitoneal route for comparison. During long-term immunization with both low and high antigen doses in the withers region, but not via the intraperitoneal route, we observed a significant accumulation of B220-CD1d-CD5-CD19+ B-2 extrafollicular plasmablasts in the subcutaneous fat and regional lymph nodes but not in the intraperitoneal fat. Only low antigen doses induced a significant accumulation of CXCR4+ CXCR5- CD4+ extrafollicular T helpers in the withers adipose tissue but not in the regional lymph nodes or abdominal fat. Only in subcutaneous fat was there a combination of extrafollicular helper accumulation. In conclusion, extrafollicular B- and T-cell activation are necessary for early IgE class switching.

## 1. Introduction

It is generally accepted that the development of allergic immune responses in individuals depends on the properties of barrier tissues, including lymphoid structures in their vicinity [1,2,3]. Certain aspects of gene structure and expression levels may be responsible for poor tight-junction formation and a predisposition to barrier-tissue damage [1], which, in turn, leads to the production of a type 2 response that includes the release of cytokines and subsequent T-helper-cell polarization and IgE production [4]. The importance of local tissue-associated tertiary lymphoid structures in the allergic immune response has recently become more evident. Clinical studies of nasal polyps in patients with allergic rhinitis [5,6,7], as well as some experimental mouse asthma models [8,9,10], demonstrated that mucosa-associated B-cells could be activated for class-switch recombination in situ in nasal polyps or inducible bronchial-associated lymphoid tissues (iBALTs).

Allergens penetrate not only via the airway epithelium but also through skin underlined with subcutaneous adipose tissue, but the role of fat-associated lymphoid clusters (FALCs) or fat-associated B-cells in the allergic immune response has been less well studied than the role of nasal polyps and iBALT. Some studies have shown that subcutaneous fat, especially in obese individuals, contains immunologically active B and T cells [11,12]. It is important to mention that obesity is one of the lifestyle factors linked with asthma [13]. These observations indicate that subcutaneous adipose tissues are mainly linked with obesity, and fat-associated B-cells can be responsible for initiating a pro-allergic immune responses.

The question of which mechanisms trigger IgE production and accumulation in allergic patients remains unanswered. Along with other researchers [14,15,16], we have shown that specific IgE production in allergic patients is not linked with specific IgG1 or IgG4 production, at least not in the case of non-replicative allergens. It is well known that a strong IgG response requires strong GC induction, mostly in secondary lymphoid organs. Therefore, one could suppose that IgE production is triggered at a site different from the one in secondary lymphoid organ B-cell follicles and without significant GC induction.

In most currently used allergy models, high doses of antigens are administered to mice together with adjuvants [17,18,19] to induce robust GC formation. By contrast, a low-dose IgE-inducing strategy better reflects the natural sensitization process [20,21,22]. In our previous work, we showed that low rather than high chronically administered antigen doses induced significant IgE responses with minimal IgG production after subcutaneous (SC) immunization in the withers region but not via the intraperitoneal (IP) route [23]. The B- and T-cell subpopulations in many allergy and asthma models have been thoroughly studied, but most authors have focused on the late phases of the immune response [24]; However, there is no consensus about the exact site of the cell subpopulations responsible for early IgE production.

Some research teams came to the conclusion that early IgE production occurred in early germinal centers despite the fact that, in mature germinal centers, it is greatly suppressed [25,26]. This is probably due to the inhibitory action of IL-21 [27] and the transcription repressor Bcl6 [28], which is a master regulator of germinal-center reactions [29] and an inhibitor of IgE class switching [27,28]. Importantly, the IgE+ germinal-center B-cells are relatively short-lived and do not induce long-lived plasma cells [25,26]. In his new review, C.D.C. Allen suggests that IgE class switching is favored in germinal centers when high IL-4 and low IL-21 levels are present [30]. This may be the case for early germinal centers when follicular T helpers are not fully mature and secrete low amounts of IL-21, but active T-helper-2 cells in T-cell zones near B-cell follicles are already present. However, Allen also suggests that, in the tissue microenvironment, a low local concentration of IL-21 may also be favorable for IgE class switching [30]. Some recent studies showed that, especially in local sites such as nasal polyps, IgE class switching may occur extrafollicularly [31] or in the absence of detectable germinal-center induction [7] despite the fact that others showed that IgE+ B cells may have a direct germinal-center origin [32,33]. Overall, the question about the exact site and cell subpopulations responsible for early IgE production remains unanswered; thus, in this work, we investigated the early stages of allergen-specific immune responses to estimate the B-cell subpopulations responsible for early IgE B-cell class switching in subcutaneous fat compared to regional lymph nodes.

Previously, we showed that early IgE production in mice mostly occurred after long-term antigen administration in the withers region containing subcutaneous adipose tissue with numerous fat-associated lymphoid clusters (FALCs). By contrast, the role of regional lymph nodes in the early stages was relatively limited. IgE production was also delayed in the case of intraperitoneal immunization despite the fact that abdominal fat also contains FALCs. We supposed that these differences could be due to different B-cell subpopulations being activated during long-term antigen administration in withers subcutaneous fat compared to regional lymph nodes and abdominal adipose tissue. It is also likely that the differences in immune response for the low and high antigen doses were also due to the different immune cell subpopulations activated in each case. To verify this hypothesis, we analyzed the contents of different B-cell subpopulations in these sites at different time points during long-term antigen administration. We also investigated the role of certain T-cell subpopulations. A deeper understanding of these mechanisms will help improve current strategies for allergy and asthma prevention in predisposed individuals.

## 2. Materials and Methods

### 2.1. Mice

All animal experiments were carried out according to the IBCh RAS IACUUC protocol. Female BALB/c mice (6–8 weeks) were obtained from the Andreevka Center (Stolbovaya, Russia). For 2 weeks, the mice were housed in plastic cages, with 10–12 per cage, under conventional minimal pathogen conditions. They were kept in a 12 h light/dark cycle at room temperature and fed ad libitum.

### 2.2. Immunization, Allergen Challenge, and Sample Collection

The mice received OVA (>98% Purity, Art. A5503, Sigma Aldrich, Darmstadt, Germany) as a model antigen 3 times a week for 4 weeks (28 days). A low (100 ng) or high (10 µg) dose of OVA per injection was administered subcutaneously the in withers region or by the intraperitoneal route (IP), but only in low doses. An endotoxin measurement demonstrated that, with low OVA doses, lipopolysaccharides were undetectable, while with high doses, 0.04 EU/dose was present.

An antigen was administered in 100 µL of sterile saline. Intact mice or saline-treated animals were used as control groups. There were 20 mice in each experimental group, and every 7 days, 5 mice from each of the three experimental groups were challenged with 0.2 mL of 0.25% OVA solution to estimate anaphylaxis severity. Body temperature was measured by using an infrared thermometer, CEM DT-8806S (SEM Test Instruments, Moscow, Russia) [34]. The temperature was taken every 15 min for 1.5 h. We observed that the most significant temperature decline was detected after 45 min, and the magnitude of this decline was considered to be a quantitative indicator of anaphylaxis severity. The magnitude of the decline was never more than 2.5 °C in the case of animal survival. In some cases, however, we observed death 30–60 min after challenge. In these cases, the severity of the anaphylaxis was believed to be higher than that in the surviving mice. Therefore, we assigned values to different death time points (–dTs): “3” for 1 h, “4” for 45 min, and “5” for 30 min after challenge.

Blood was taken retro-orbitally from the anesthetized animals. Serum was collected by centrifugation and stored at −20 °C before use. The mice were sacrificed by isoflurane (“Aeran”, Baxter, Compton, UK) inhalation. During the preparation of samples from subcutaneous fat, the mice were perfused with 3 mL of PBS with a heparin solution via the retro-orbital sinus before killing. Withers or abdominal adipose tissue samples and axillary lymph nodes were collected. Quantitative PCR samples were homogenized in ExtractRNA (Evrogen, Moscow, Russia). For flow cytometry, homogenization was performed in PBS (pH = 7.2). Fat homogenates of lymphocytes were centrifuged (300× *g*) and washed twice with PBS. Lymph-node lymphocytes (5 × 10^5^ cells) were resuspended in ExtractRNA for gene expression measurement. The remaining cells were taken for flow cytometry.

### 2.3. ELISA for OVA-Specific Antibody Assay

ELISA for the detection of specific IgE production was carried out on 96-well microtiter plates (Costar, Thermo Scientific, Waltham, MA, USA) coated with 50 mL of 20 µg/mL OVA solution in PBS (pH = 7.2) overnight at 4 °C. Between each stage, the plates were washed 3–4 times with 0.05% Tween-20 (PBS-T) by using an automatic washer, PP2 428 (“Immedtech”, Dubna, Russia). After overnight incubation and subsequent washing, the plates were blocked with 5% BSA in PBS for 1 h at room temperature (100 µL in each well). At the next stage, the plates were incubated with different serum dilutions in the same blocking buffer overnight at 4 °C (50 µL in each well). After subsequent washing, we used direct HRP-labeled anti-mouse IgE antibodies. To detect specific IgE, we used anti-mouse IgE-HRP (clone 23G3) at a 1:1000 dilution in a blocking buffer (50 µL in each well). The plates were incubated with this conjugate for 3 h at room temperature. They were then washed again with PBS-T, after which 50 µL of the highly sensitive 3,3′5,5′-tetramethylbenzidine (TMB) substrate (ab 171523, Abcam, Cambridge, UK) was added to each well, before incubation for 30 min. The optical densities (ODs) were measured by using an automatic plate reader (Thermo Fisher Scientific, Waltham, MA, USA) at 450 nm, with the subtraction of the optical density at 620 nm. The antibody quantities were estimated as the serum titers corresponding to the maximal serum dilutions at which the ODs were three standard deviations higher than the mean background ODs.

The ELISA protocol for specific IgG1 production was slightly modified. Coating was performed by using a 5 µg/mL OVA solution in PBS (pH = 7.2). Blocking was performed by using 1% BSA in PBS. The serum samples and conjugates were also added in 1% BSA in PBS. Following the incubation of the serum samples, the plates were further processed with anti-mouse IgG1 (clone RMG1-1) or anti-mouse IgG3 (clone RMG3-1) at a 1:1000 dilution (BioLegend, San Diego, CA, USA) for 2 h at room temperature.

### 2.4. Gene Expression Measurement

RNA was extracted by using the standard phenol–chloroform method, followed by RNAse-free DNAse treatment (Thermo Fisher Scientific, Waltham, MA, USA). Briefly, small patches of subcutaneous fat tissue (about 3 × 3 × 3 mm) from each mouse were homogenized in 1 mL of commercial ExtractRNA solution (Evrogen, Moscow, Russia) containing phenol. For the lymph nodes, about 3 × 10^5^ cells in 20 µL of PBS were added to 1 mL of ExtractRNA. The samples were incubated for 30 min at room temperature and then kept at −20 °C prior to RNA purification. About 0.25 mL of chloroform was added to 1 mL of the sample during RNA extraction. The samples were vortexed for 3 min at room temperature, after which they were centrifugated at 15,000× *g* for 15 min at 4 °C. The upper aqueous phase was transferred to a new 1.5 mL tube, and then 600 µL of isopropanol was added to the sample. The samples were incubated for 30 min at −20 °C for precipitation and then centrifuged for 10 min at 12,000× *g*. After supernatant decantation, 1 mL of 75% ethanol was carefully added to each sample, after which all the samples were centrifuged again for 10 min at 12,000× *g*. The supernatant was discarded, and the pellets were washed with ethanol again. After that, the samples were heated at 50 °C in uncapped 1.5 mL tubes for 3–5 min to dry up the remaining ethanol.

To remove genomic DNA from our samples, we used RNAse-free DNAse I (EN 0525, Thermo Scientific, Waltham, MA, USA) according to the manufacturer’s protocol. Samples were diluted in 8 µL of DEPC-treated water with the addition of 1 µL of 10× reaction buffer and 1 µL of DNAse I (1U) solution. Incubation at 37 °C for 30 min was performed. To activate DNAse I without RNA degradation occurring in the presence of divalent cations, we added 1 µL of 50 mM EDTA and incubated the samples at 65 °C for 10 min. The remaining template was diluted to 300 µL in DEPC water and kept at −20 °C before usage.

For the measurement of DNA excision circles corresponding to direct and sequential IgE switches, we did not perform DNA digestion; rather, cDNA was synthesized using a RevertAid First Strand cDNA Synthesis Kit (Thermo Fisher Scientific). A quantitative PCR was performed using kits from BioLabMix (Novosibirsk, Russia). Probes with 6-FAM as a fluorescent dye on the 5′-end and BHQ-1 as a quencher on the 3′-end were used. The expression of target genes and the presence of excision DNA circles were estimated by normalizing to the expression of 2 housekeeping genes, GAPDH and HPRT, and were calculated as 2^−^^Δ(^^ΔCt)^ compared to the expression in the tissues of intact mice. The data are shown as the relative expression determined as the ratio of 2^−^^Δ(^^ΔCt)^ for the experimental groups to that for the intact mice. The reaction was performed in a CFX Connect Amplificator (BioRad, Hercules, CA, USA) according to the following protocol: 95 °C initial denaturation for 3 min followed by 50 cycles; 5 s denaturation at 95 °C; 20 s annealing and elongation at 64 °C. The reaction was performed in 96-well plates (MLP9601, BioRad Hercules, CA, USA) in a 20 µL volume. The forward and reverse primer concentrations were 0.4 µM each, and the probe concentration was 0.2 µM. About 3 µL of diluted template was used in each reaction. The primers and probes were designed in NIH Primer BLAST and synthesized by Evrogen (Moscow, Russia).

The following primers and probes were used in this study:

GAPDH F: GGAGAGTGTTTCCTCGTCCC; R: ACTGTGCCGTTGAATTTGCC;

Z: /6-FAM/- CGCCTGGTCACCAGGGCTGCCATTTGCAGT-/BHQ-1/;

HPRT F: CAGTCCCAGCGTCGTGATTA; R: TCCAGCAGGTCAGCAAAGAA;

Z: /6-FAM/- TGGGAGGCCATCACATTGTGGCCCTCTGTGTG /BHQ-1/;

germline ε F: CCCACTTTTAGCTGAGGGCA; R: CTGGTTAAGGGCAGCTGTGA;

Z: /6-FAM/- CGCCTGGGAGCCTGCACAGGGGGC-/BHQ-1/;

germline γ1 F: AGAACCAAGGAAGCTGAGCC; R: AGTTTGGGCAGCAGATCCAG;

Z: /6-FAM/- AGGGGAGTGGGCGGGGAGGCCA-/BHQ-1/;

circular µ-ε F: CCCACTTTTAGCTGAGGGCA; R: CGAGGGGGAAGACATTTGGG;

Z: /6-FAM/- CGCCTGGGAGCCTGCACAGGGGGC-/BHQ-1/;

circular γ1-ε F: AGATTCACAACGCCTGGGAG; R: GTCACTGTCACTGGCTCAGG;

Z: /6-FAM/- CCACTGGCCCCTGGATCTGCTGCCCA-/BHQ-1/;

postswitch ε F: CCAGTCCACATGCTCTGTGT; R: AGCGTGGGGAACTGGTTAAG;

Z: /6-FAM/- TGGGGTCCCCAGAGCCCTGCTCCTGT-/BHQ-1/;

postswitch γ1 F: CCTCTGGCCCTGCTTATTGT; R: GTCACTGTCACTGGCTCAGG;

Z: /6-FAM/- CCACTGGCCCCTGGATCTGCTGCCCA-/BHQ-1/;

AICDA F: CCTCTGCTACGTGGTGAAGA; R: GCTGAGGTTAGGGTTCCATCT;

Z: /6-FAM/- CTGGAGCCCGTGCTATGACTGTGCCCGGCA-/BHQ-1/.

### 2.5. Flow Cytometry

To prepare the cell for flow cytometry, patches of subcutaneous white adipose tissue as well as regional lymph nodes were meticulously homogenized in PBS (pH = 7.2). Before homogenization, the tissue patches were incubated for 30 min at 37 °C with a solution containing 1 mg/mL collagenase I and 75 U/mL DNAse I (Sigma Aldrich, Darmstadt, Germany). The cells were washed in PBS and pelleted by centrifugation twice at 300× *g*. Cells from adipose tissue homogenates, and lymph nodes were passed through an 80 µm mesh filter to obtain a single-cell population. The cells were washed in PBS and stained with antibodies for their respective markers. To discriminate B-cell subpopulations, antibodies were used as follows: CD5-BV510 (clone 53–7.3, BioLegend), CD1d-FITC (clone 1B1, BioLegend), CD95-PE (clone SA367H8, BioLegend), CD38-PECy7 (clone 90, BioLegend), CD19-APC (clone 6D5, BioLegend), and B220-APCCy7 (clone RA3-6B2, BioLegend). To discriminate the ILC2-cell, NK-cell, and T-cell subpopulations, organ homogenates were stained: CD4-BV510 (clone GK1.5, BioLegend), CD49b-PE (clone HMa2, BioLegend), CXCR4-PerCPCy5.5 (clone L276F12, BioLegend), CXCR5-PECy7 (clone L138D7, BioLegend), ST2-APC (clone DIH4, BioLegend), CD45-APCCy7 (clone 30F11, BioLegend), and a biotinylated anti-lineage cocktail (BioLegend, cat # 133307), followed by streptavidin–FITC (Biolegend). For the isotype control, we used anti-rat IgG1 antibodies (clone GO114F17) labeled with BV510, FITC, PE, PerCPCy5.5, PECy7, APC, and APCCy7, respectively.

After pre-blocking with 10% rabbit serum for 15 min, the cells were stained for 1 h at 4 °C in a FACS buffer (0.5% BSA, 0.01% NaN_3_ in PBS, pH = 7.2). To exclude dead cells, DAPI at 0.1 µg/mL was added 15 min prior to flow cytometry.

B-1a cells were identified as CD19 + B220-CD5+, MZ-B cells as CD19 + B220 + CD1d+ [35], GCs as CD19 + B220 + CD38-CD95+ [36], and extrafollicular plasmablasts as CD19 + B220- [37]. The different expression levels of CD38 and CD95 made it possible to discriminate these populations. Follicular T helpers were identified as CD45+ Lin+ CD4+ CXCR4+ CXCR5+, extrafollicular T helpers as CD45+ Lin+ CD4+ CXCR4+ CXCR5- based on observations from [38], and Th2 as CD45+ Lin+ CD4+ CXCR4-CXCR5-ST2+ cells [39].

Flow cytometry was performed on a MACS Quant Tyto (Miltenyi Biotech, Gladbach, Germany). The results were processed in FlowJo v.10 (BD, Columbus, OH, USA).

### 2.6. Histology

Withers tissue was taken from isoflurane-sacrificed, low-dose-immunized mice on Day 28. Samples were fixed with 4% PFA and kept in the 4% PFA before the preparation of the histological sections. The tissue samples were dehydrated in solutions with increasing ethanol concentrations (70, 80, 95, and 100%) for 45 min and then with xylene for 1 h. The tissue samples were then embedded in paraffin. Histological sections that were 8 nm thick were made on a microtome (Thermo Fisher Scientific HM 355S, Thermo Fisher, Waltham, MA, USA). The histological sections were rehydrated by incubating them twice for 5 min in a xylene solution and then twice in 100% ethanol for 5 min, followed by two incubations each in 95% ethanol and 70% ethanol for 5 min and, finally, two incubations in water. Staining was performed using a hematoxylin and eosin staining kit (Abcam, Cat # ab245880, Cambridge, UK) or Picro Sirius Red (Abcam, Cat # ab150681) according to the manufacturer’s instructions. Briefly, in the case of hematoxylin and eosin staining, slides were incubated for 5 min in Mayer’s hematoxylin, followed by rinsing in water, bluing reagent, and 100% ethanol. After that, incubation with Eosin Y solution for 3 min was performed, followed by rinsing in 100% ethanol. In the case of Picro Sirius Red, the slides were incubated for 1 h with Picro Sirius Red solution, followed by washing with acetic acid and 100% ethanol.

### 2.7. Statistics

All the experiments were performed 2–3 times. To compare the experimental groups, an ANOVA test with correction by multiple comparisons was used. *p* < 0.05 was considered statistically significant. To determine the correlation coefficients, Spearman’s test was used. The means and standard deviations for each compared group were calculated.

## 3. Results

### 3.1. Chronically Low-Dose Adjuvant-Free Antigen Administration in Subcutaneous but Not Abdominal Fat Tissue Induced Early B-Cell IgE Class Switching and Reproduced IgE-Mediated Type I Hypersensitivity

We previously showed [23] that long-term antigen administration induced highly specific IgE titers mainly when the antigen was administered in the withers region by the SC route. Indeed, we observed that a low (100 ng) dose of OVA induced a substantial IgE response from the 14th day of the immunization protocol. This response reached a plateau on the 21st day, along with a specific IgG1 response (Figure 1a,b). At the same time point, mice developed a pro-anaphylactic immune reaction following a temperature drop after a high-dose allergen challenge (Figure 1c). We also observed specific IgE production after a high (10 µg) dose in immunized animals. The IgE levels were comparable between the low and high dose groups by the 28th day (Figure 1a). In the high-dose group, IgE production reached such a level only on the 28th day and was accompanied by very high specific IgG1 production (Figure 1a,b). In high-dose-immunized animals, anaphylactic severity was greater than that in their low-dose-immunized counterparts (Figure 1c), and an allergen challenge provoked high mortality in the high-dose group. Despite this, there was no significant correlation between IgE production and anaphylactic severity, contrary to the results for the low-dose group (Figure 1g,h).

Therefore, in this case, anaphylaxis could have been due to the presence of pro-anaphylactic IgG1 antibodies. These IgG1 antibodies in some cases trigger mast-cell degranulation [40] or the release of platelet activation factor from macrophages [41]. Neither pathway was observed to be activated during classical human type I allergy responses [42]. By contrast, we observed a significant correlation between the IgE production and anaphylaxis severity in the low-dose group (Figure 1g). As demonstrated, the chronic administration of both low and high antigen doses was characterized by comparable specific IgE production. At the same time, the ratio of IgE to IgG1 in the low group was 1:10, while in the high-dose one, it was 1:1000 or higher (Figure 1). Therefore, the low-dose allergy model better reproduced the clinical pathogenesis of sensitization and IgE-mediated type I allergy development in humans.

By contrast, when low antigen doses were administered by the IP route in a region containing visceral fat with visceral FALCs, IgE production appeared only 4 weeks after beginning of the protocol and was about 10 times lower than that after immunization in the withers region at the same time point. Specific IgG1 production was also delayed, and a significant pro-anaphylactic response was observed only 4 weeks after immunization (Figure 1d–f).

### 3.2. Subcutaneous-Fat-Associated B Cells Are Responsible for Early B-Cell IgE Class Switching

Next, we addressed the question of whether B-cell IgE class switching occurred exclusively in subcutaneous-tissue-associated B-cells and if they formed FALCs. In our previous work [23], we showed that B-cell IgE isotype switching, but not IgE production, was triggered in tissue-associated B cells. In this study, we used a more accurate method to quantify gene expression—probe-based quantitative PCR tests—instead of the SYBR Green I-based technique. Indeed, in accordance with our previous results, we showed that, at early time points, from the 7th to the 21st day of immunization, the induction of germline transcripts linked with isotype switching occurred almost exclusively in the withers adipose tissue (Figure 2a,c) but not in regional lymph nodes (Figure 2b,d). Meanwhile, the circular µ-ε DNA excision circles were detected in the lymph nodes of high- and low-dose mice on the 14th and 21st days, respectively (Figure 2b,d). These excision circles could have originated from B cells that had recently migrated to the lymph nodes from the isotype-switching site. It is, therefore, more likely that, during the first 3 weeks of chronic antigen administration, B-cell IgE isotype switching occurred exclusively in subcutaneous-fat-associated B-cells. Moreover, we could not observe the stable induction of the AICDA gene, which encodes enzyme AID, which is responsible for isotype switching, in lymph nodes, in addition to a weak induction after low-dose immunization at one early time point (Figure 2b,d). By contrast, the induction of AICDA expression in subcutaneous fat began at earlier time points (7th–14th days) and continued throughout the protocol (Figure 2a,c). The accumulation of post-switch ε and post-switch γ1 transcripts was also observed mainly in subcutaneous fat (Figure 2a–d). 

However, in contrast to our previous results [23], we also observed the expression of markers associated with B-cell IgE class switching in the regional lymph nodes after 4 weeks of antigen administration (Figure 2b,d). Germline ε transcripts in withers adipose tissue were induced at early time points before the 21st day of immunization. In the regional lymph nodes, this induction occurred only on the 28th day.

This indicated that only after a long time do low antigen doses accumulate in a sufficient quantity not only at the sites of the antigen administration but also in regional lymph nodes. Another possibility is that the antigen reached the lymph nodes during the first week of administration, but certain specific but unidentified niche factors suppressed early B-cell IgE and IgG1 switching in secondary lymphoid organs (SLOs).

The induction of stable AICDA expression was observed in visceral abdominal fat after IP immunization from an early time point. In contrast, the induction of germline and circular transcripts corresponding to IgE class switching was not detected until the 21st day. Therefore, in abdominal fat, despite B-cell activation in general, IgE class switching was hampered probably because of the lack of certain specific factors present in subcutaneous fat.

### 3.3. Subcutaneous Fat in Mouse Withers Region Contains Organized Tertiary Lymphoid Structures

It is well known that visceral adipose tissue contains a large number of FALCs [43], which resemble structures found in human subcutaneous adipose tissue [11]. To visualize these structures in the present study, we performed H and E (Figure 3a,c–e) and Picro Sirius collagen I and III fiber histological staining (Figure 3b,f) of subcutaneous fat from the withers region of low-dose-immunized mice. Figure 3 shows the existence of large fat-associated dense FALC-like structures at the border of adipocytes. We found only one large, dense FALC-like structure for each withers tissue, and not all cells were collected within one site in all mice (Figure 3c–e). These structures were well organized and had collagenous membranes (Figure 3f).

### 3.4. B-2-Cell-Derived Plasmablasts but Not GCs Are Responsible for IgE Production after Long-Term Antigen Administration

IgE class switching in abdominal fat tissue and in regional lymph nodes close to the withers region occurs at later time points than that in subcutaneous fat tissue. This may be linked with the different B-cell subpopulations activated in these sites upon antigen administration. To verify this hypothesis, we performed flow cytometric analysis of B-cell subpopulations isolated from the withers and regional lymph nodes at different time points during long-term antigen administration in subcutaneous adipose tissue. The B-cell gating strategy is shown in Appendix A. Figure 4 also clearly shows that there was no GC B-cell (CD19+ B220+ CD38− CD95+) induction in either the subcutaneous withers adipose tissue or the regional lymph nodes. Although high doses of antigen induced the accumulation of GC B-cells, this induction was transient in the subcutaneous fat and was detected only on the 21st day. This observation indicates that the conditions in the subcutaneous fat were unfavorable for GC persistence. Instead, a significant accumulation of CD19+ B220− plasmablasts was observed (Figure 3a,d–g). Most of these CD19+ B220−plasmablasts were CD38− CD95+ or CD38+ CD95− (Figure 3b). These cells—in the subcutaneous fat but not in the regional lymph nodes—outnumbered B cells with the classical germinal center phenotype CD19+ B220+ CD38− CD95+ (Figure 4). The absence of CD38 on some of them and the presence of CD95 markers may have indicated that these cells were closely related to classical GC cells, differing only in the absence of B220 expression. The other possibility is that this phenotype simply reflected a fully activated B-cell state.

The accumulation of activated plasmablast subpopulations was not observed in abdominal fat tissue at the early time points. Some transient increase in the amount was observed on the 21st day, which coincides with the delayed induction of transcripts corresponding to IgE class switching after IP immunization (Figure 4e).

The percentage of GC B cells in the withers adipose tissue was inversely correlated with specific IgE titers. There was no functional association between the number of GC B cells in the local lymph nodes and IgE levels (Figure 5c,f). By contrast, significant correlations were observed between the percentage of CD19+ B220− CD38− CD95+ activated plasmablasts in the subcutaneous fat and regional lymph nodes and IgE titers (Figure 5a,d). As observed in the gating strategy plots (Appendix A), these plasmablast subpopulations were not derived from B-1a or MZ-B B-cells. For CD19+ B220− CD38+ CD95+ Plasmablasts, this correlation was significant only for regional lymph nodes (Figure 5b,e). Despite the fact that some plasmablasts were CD38+ CD95− (naïve-like) or even CD38− CD95− (see Figure 4b), the percentages of these minor subpopulations did not correlate with the IgE titers (data not shown).

In comparison to the B-2-derived plasmablasts, no increase in T-cell-independent B-1a or MZ-B B cells was detected before the 21st day after low-dose immunization (data not shown). It is very unlikely that these cells were responsible for the early IgE class switching after low-dose immunization.

We observed that the percentages of different plasmablast subpopulations in the regional lymph nodes and the withers tissue started to increase at the same time points. This meant that the antigen, even at low doses, was rapidly delivered to the regional lymph nodes. Thus, one could suppose that extrafollicular plasmablast accumulation per se is necessary but not sufficient for IgE production. The impact of different types of T helpers on these cells in the withers adipose tissue on the one hand and the regional lymph nodes and in the visceral fat on the other could result in delayed IgE switching in regional lymph nodes and visceral fat tissue.

### 3.5. Extrafollicular T-Helper-Cell Accumulation Results in High IgE Production Accompanied by Minimal Igg1 Production in Response to Low Antigen Doses

Different T-helper subpopulations could be responsible for the specific humoral response pattern in response to a low antigen dose. The different activity of these cells in withers tissue and regional lymph nodes could account for a delayed IgE isotype switching in regional lymph nodes compared to in subcutaneous adipose tissue B cells.

The gating strategy for the T-helper-cell subsets is shown in Appendix A. Presuming that extrafollicular proliferating plasmablasts account for specific IgE production, it is logical that this production should also be directly linked with extrafollicular T helpers that are specific for supporting extrafollicular plasmablasts [43]. IgE production is reciprocally linked with follicular T helpers that support GC function and suppress the exit of B cells from the GC by stimulating Bcl-6 expression [44]. Indeed, we observed a remarkable decrease in CXCR4+ CXCR5+ GC follicular T helpers in the subcutaneous fat tissue on the 21st day when IgE response had plateaued. A decrease in this T-helper-cell subpopulation could account for GC destabilization and the competitive enhanced development of extrafollicular plasmablasts (Figure 6b). By contrast, CXCR4+ CXCR5- extrafollicular T helpers only accumulated transiently in the subcutaneous fat in the low-dose-immunized mice on the 21st day (Figure 6a,b).

There was no accumulation of these cells in the regional lymph nodes. At the same time, CXCR4- CXCR5+ T helpers accumulated significantly in the high-dose group in particular (Figure 6c–e). We suggest that CXCR4- CXCR5+ CD4+ cells could be T helpers residing in B-cell follicles outside the GC and that these cells could also support B-follicle structure [38]. The accumulation of the GC-supporting CXCR4+ CXCR5+ follicular T helpers was also observed in the regional lymph nodes on the 21st day in the high-dose group (Figure 6e).

Upon antigen administration by the IP route, a significant although transient accumulation of CXCR4+ CXCR5+ follicular T helpers at an early time point and a significant decline in the CXCR4+ CXCR5- extrafollicular T-helper percentage were detected (Figure 6f). This was markedly different from the situation observed after the administration of antigen in the withers region. This could be the reason for the transient nature of extrafollicular B-cell response in visceral fat tissues.

Our gating strategy (Appendix A) also allowed us to determine the potential accumulation of NK cells (CD45+ CD49b+). However, at no time point did we observe a statistically significant accumulation (data not shown); thus, these cells did not factor in our case for the early IgE class switching and antibody production.

Consequently, a decrease in CXCR4+ CXCR5+ and an absence of accumulated CXCR4- CXCR5+ T helpers, among the CD4+ T-cells, stabilizing the GC and B-cell follicles, respectively, resulted in accelerated specific IgE production in the subcutaneous fat. CXCR4+ CXCR5- extrafollicular T-helper accumulation in the low-dose group supported specific IgE but not IgG1 production after long-term antigen administration at high levels compared to in the high-dose-immunized mice. The early although transient increase in follicular T-helper accumulation, along with the later decrease in the extrafollicular T-helper percentage in visceral abdominal fat, was the likely reason for the unstable extrafollicular B-cell response and delayed IgE production after antigen administration by the IP route compared to administration in the withers region by the SC route.

## 4. Discussion

In this study, we found the primary site of the antigen-induced B-cell IgE isotype switching in the low-dose allergen model and investigated the main patterns of B- and T-cell activation. In our allergy model, the route of immunization and allergen dose played essential roles. Although long-term immunization with both low and high antigen doses induced comparable levels of IgE production, the ratio of IgE to IgG1 in the low-dose groups was 1:10, while it was 1:1000 or higher in the high-dose ones. Consequently, the low-dose immunization protocol better reproduces clinical findings. Moreover, only with the low-dose administration protocol did the anaphylaxis severity correlate with specific IgE production, which resembles clinical manifestations in humans [14,15,16]. Our data are in agreement with previous research reporting that the long-term adjuvant-free low-dose allergy model better reflected the natural sensitization process [20,21,22]. Both subcutaneous-fat-associated and local lymph node B cells were activated upon adjuvant-free long-term antigen administration, and they produced specific IgE antibodies. Subcutaneous fat was the primary site of antigen-induced IgE class switching.

According to the classical point of view, B-cell class switching and the generation of high-affinity antibodies to T-cell-dependent antigens occur in germinal centers [29]. Recently, however, it was shown that, at least in some cases, class switch recombination occurs after B-cell activation prior to the formation of germinal centers or extrafollicular foci, whereas only somatic hypermutation occurs within germinal centers [45]. However, we did not observe accumulation of B cells with the germinal-center phenotype upon low-dose antigen administration. By contrast, the accumulation of CD19+ B220- extrafollicular B-cell plasmablasts was clearly observed in both subcutaneous fat and regional lymph nodes. Conventional plasmablasts are not believed to form GC-like structures. However, in the present study; we showed that some B220- plasmablasts acquired a GC-like phenotype (CD38- CD95+). The percentages of these cells both in subcutaneous fat tissue and regional lymph nodes significantly correlated with the IgE titers in individual mice; thus, it is likely that these cells were mostly engaged in specific IgE production. This phenomenon could result from the interaction of these cells with T helpers that stimulate CD95 expression and downregulate CD38 expression on B cells via CD40 ligation [46]. The significant correlation between the percentage of these CD38- CD95+ plasmablasts and IgE titers may indicate that these cells corresponded to the final stages of the differentiation of IgE-producing cells. If B220- CD38- CD95+ plasmablasts are cells that contact T cells during the response to low antigen doses, it can be assumed that the presence of such a T-helper subpopulation is crucial for the development of IgE responses. To investigate the participations of follicular T helpers and extrafollicular T helpers in our model, we measured these populations by flow cytometry. We showed that the accumulation of extrafollicular T-helper cells without the accumulation of T helpers situated in B-cell follicles mostly occurred in the subcutaneous fat tissue of low-dose-immunized mice only. Moreover, 3–4 weeks after antigen administration into visceral abdominal fat tissue, we observed a decrease in the percentage of extrafollicular T-helper cells. This seems to be a plausible reason for the transitory but unstable accumulation of B220- CD38- CD95+ plasmablasts in these sites. In our study, we showed that B-cell IgE class switching and early IgE production were mostly linked with extrafollicular B-cell activation. This was in agreement with some literature data because several studies suggested that extrafollicular B-cell class switch recombination was possible at least at the early stages of plasmablast differentiation [45,47,48]. Furthermore, recent work clearly showed that B-cell class switch recombination, in some cases, occurs mostly at the early stages of B-cell activation before differentiation into GC centroblasts or extrafollicular B-cell blasts and is mostly dependent on T–B-cell contacts per se but not on GC formation [45].

One of the most important results from our work is that, in persons predisposed to allergies induced by low antigen doses, penetration through the mucosal barrier is markedly different in subcutaneous vs. abdominal fat. Numerous studies have aimed to elucidate the mechanisms of the formation and function of fat-associated lymphoid clusters in visceral abdominal adipose tissue [43]. Despite the fact that allergens penetrating the skin barrier enter the body through subcutaneous fat, only a few studies [11,12] have provided information on fat-associated B cells. Frasca et al. [12] observed that pro-inflammatory cytokines secreted by adipocytes promoted T-bet and CD11c expression on subcutaneous withers-associated B-cells. The expression of these molecules on CD19+ B cells identified extrafollicular plasmablast subpopulations with specific properties that are in agreement with the data obtained from this work.

## 5. Conclusions

Overall, we showed that, in subcutaneous adipose tissues, the response to low antigen doses is extrafollicular by nature and primarily based on the activation of CD19+ B220- CD38- CD95+ plasmablasts. The accumulation of CXCR4+ CXCR5- extrafollicular T helpers is also critical for their formation. This unique type of immune response creates a specific humoral pattern characterized by high IgE production accompanied by minimal IgG1 production, leading to allergy development. Subcutaneous fat tissue, but not abdominal fat and SLOs, provides the most favorable conditions for the development of such a response.

## Figures and Tables

**Figure 1 vaccines-10-00969-f001:**
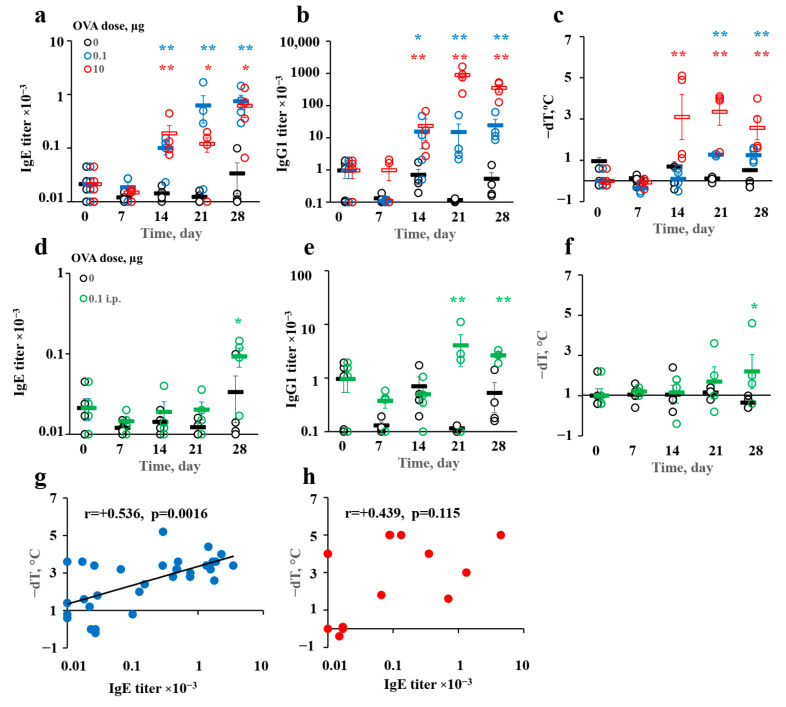
Specific IgE production was induced rapidly after immunization by the SC route in the withers region but not after IP immunization. It was accompanied by relatively low IgG1 production and correlated with systemic anaphylaxis after the administration of low antigen doses. BALB/c mice (*n* = 4 for each group–time point) were immunized by the SC route in the withers region (**a**–**c**), (**g**,**h**) or by the IP route (**d**–**f**), with indicated OVA doses for 4 weeks 3 times a week. Specific IgE (**a**,**d**) and IgG1 (**b**,**e**) production was measured at the indicated time points, as was the systemic anaphylaxis intensity 45 min after the administration of the resolving antigen dose (250 µg) (**c**,**f**). The correlation of the IgE titers with systemic anaphylaxis (**g**,**h**). Blue asterisks signify the differences between low-dose-SC-immunized mice and the intact group (*/** signify *p* values < 0.05/0.01 correspondingly); red ones signify the differences between high-dose-SC-immunized mice and the intact group; green ones signify the differences between low-dose-IP-immunized mice and the intact group. All the experiments were performed 3 times, and representative data are shown.

**Figure 2 vaccines-10-00969-f002:**
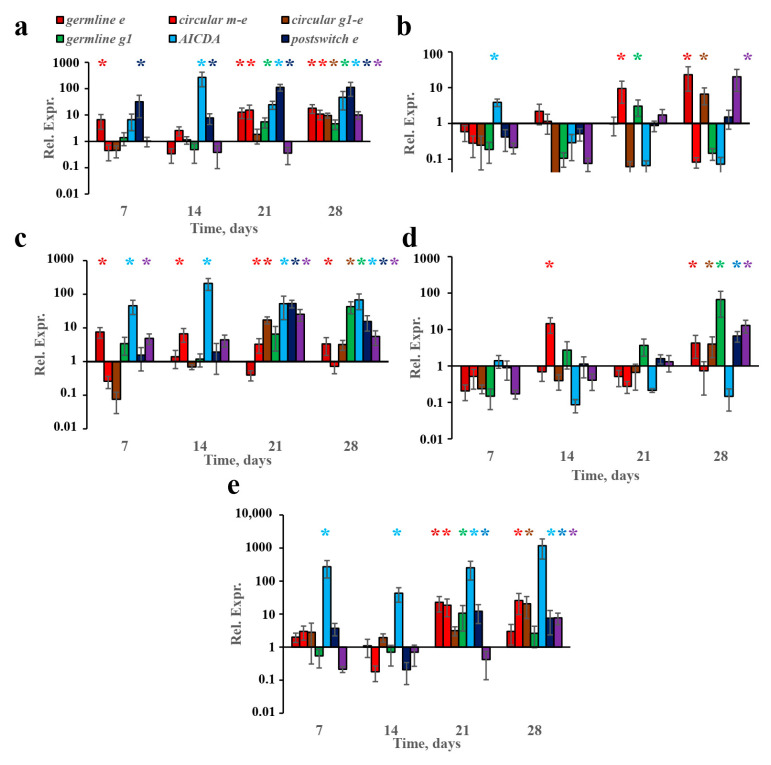
Antigen administration induced Ig class switching more rapidly in withers fat tissue compared to lymph nodes or abdominal fat. BALB/c mice (*n* = 4 for each group–time point) were immunized by the SC route in the withers region (**a**–**d**) or by the IP route (**e**) with the indicated OVA doses for 4 weeks, 3 times a week. The expression of the indicated genes and transcripts in the indicated time points are shown for low- (**a**,**b,e**) and high-antigen-dose (**c**,**d**)-immunized mice in withers subcutaneous fat (**a**,**c**), regional lymph nodes close to the withers region (**b**,**d**), and abdominal fat (**e**). Each asterisk indicates significant differences (*p* < 0.05) in gene expression shown by the same color with an intact group. Each asterisk of a certain color corresponds to the significance of the difference of the gene expression, which is shown by a column of the same color. All experiments were performed 3 times, and representative data are shown.

**Figure 3 vaccines-10-00969-f003:**
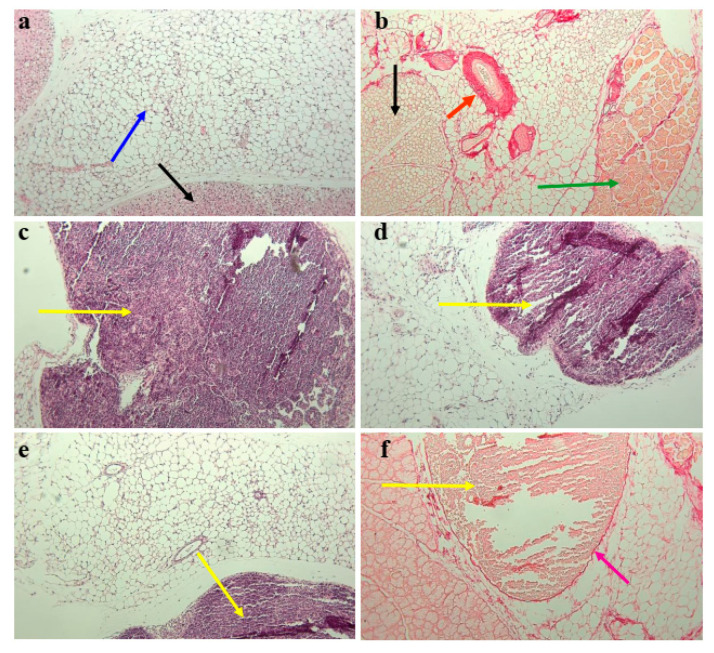
Subcutaneous fat adipose tissue in the withers region contained large bordered tertiary lymphoid structures. Representative histological images of mouse subcutaneous withers adipose tissue. Tissue samples were taken from low-dose-immunized mice 4 weeks after the start of antigen administration. Samples were stained with H and E (**a**,**c**–**e**) or Picro Sirius Red (**b**,**f**). Adipose tissue in intact mice was of two types: white, as indicated by the blue arrow in (**a**), and brown, as indicated by the black arrow in (**a**). The Picro Sirius Red-stained vessel is indicated by the red arrow in (**b**); the muscles, by the green arrow in (**b**); and the brown adipose cells, by the black arrow in (**b**). Large clusters of lymphoid cells are indicated by the yellow arrows in (**c**–**f**), stained with H and E (C–E) or Picro Sirius Red (**f**). The fibrotic border of the cluster is indicated by the purple arrow in (**f**). Magnification: 100×.

**Figure 4 vaccines-10-00969-f004:**
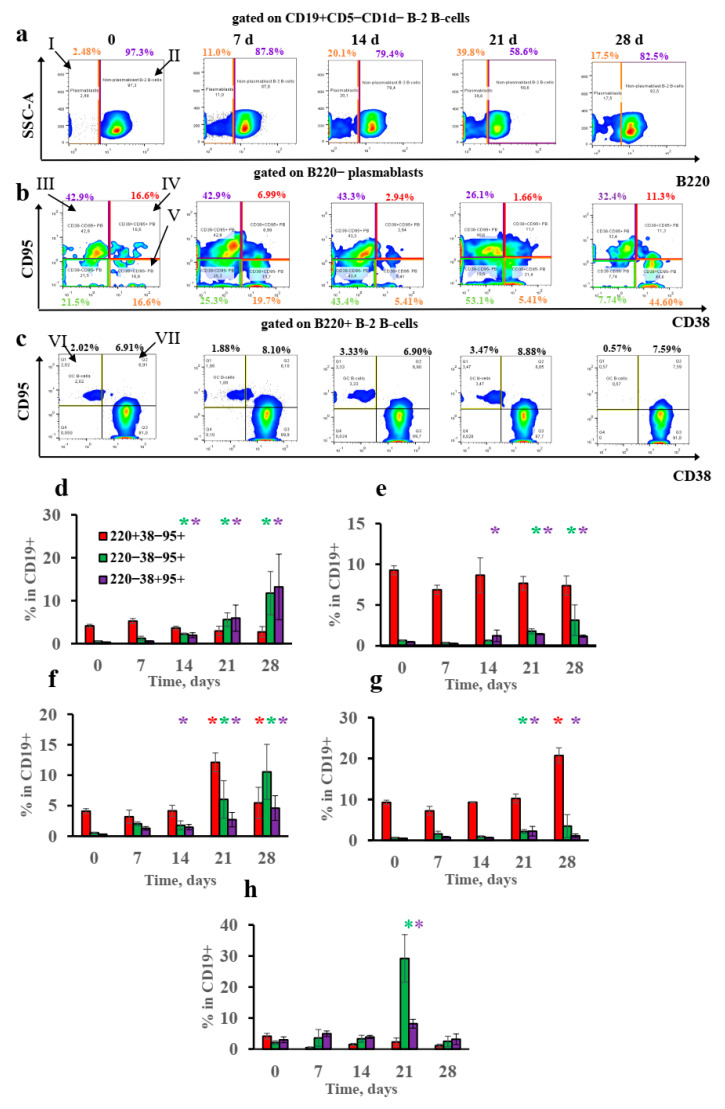
Low antigen doses triggered B220− CD38−/+ CD95+ extrafollicular plasmablasts, but not classical germinal center accumulation in withers adipose tissue and lymph nodes. BALB/c mice (*n* = 4 for each group–time point) were immunized by the SC route in the withers region (**a**–**g**) or by the IP route (**e**) with low (100 ng) or high (10,000 ng) OVA doses for 4 weeks, 3 times a week. Representative flow cytometry pseudocolor plots from mouse subcutaneous fat tissue cells gated on CD19+ Cd5- CD1d- B-2 B cells (**a**), B220− B-2 plasmablasts (**b**), and B220+ non-plasmablast B-2 B cells (**c**). Roman numerals correspond to the following subpopulations: I—B220- plasmablasts; II—B220+ non-plasmablast B-2 B cells; II–V—different plasmablast subpopulations; VI—CD38- CD95+ classical germinal centers; VII—CD38+ CD95+ activated B-2 B cells. The percentage of B220+ CD38− CD95+ classical germinal centers and B220− CD38− CD95+ and B220− CD38+ CD95+ plasmablasts at the indicated time points are shown for low (**d**,**e**,**h**)- and high-antigen dose (**e**,**g**)-immunized mice in withers subcutaneous fat (**d**,**f**), regional lymph nodes close to the withers region (**e**,**g**), and abdominal fat (**e**). Each asterisk indicates significant differences (*p* < 0.05) in the percentages of the subpopulation in B cells shown by the same color with the intact group. Each asterisk of a certain color corresponds to the significance of the difference of subpopulation percentage, which is shown by a column of the same color. All experiments were performed 3 times, and representative data are shown.

**Figure 5 vaccines-10-00969-f005:**
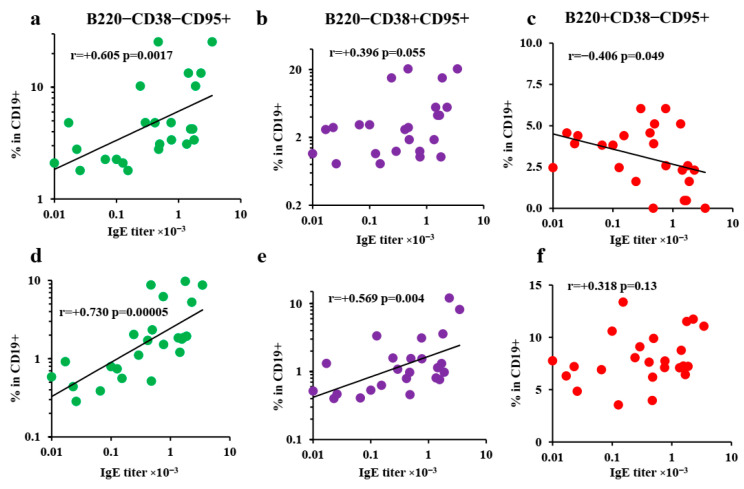
Proportions of extrafollicular plasmablast subpopulations correlated with IgE production. BALB/c mice (*n* = 4 for each group–time point) were immunized by the SC route in the withers region with the indicated OVA doses for 4 weeks, 3 times a week. Correlation of the percentage of CD38− CD95+ (**a**,**d**) and CD38+ CD95+ (**b**,**e**) extrafollicular plasmablasts in B cells and B220+ CD38− CD95+ classical germinal centers (**c**,**f**), in withers adipose fat (**a**–**c**), and regional lymph nodes (**d**–**f**). All the experiments were performed 3 times, and representative data are shown.

**Figure 6 vaccines-10-00969-f006:**
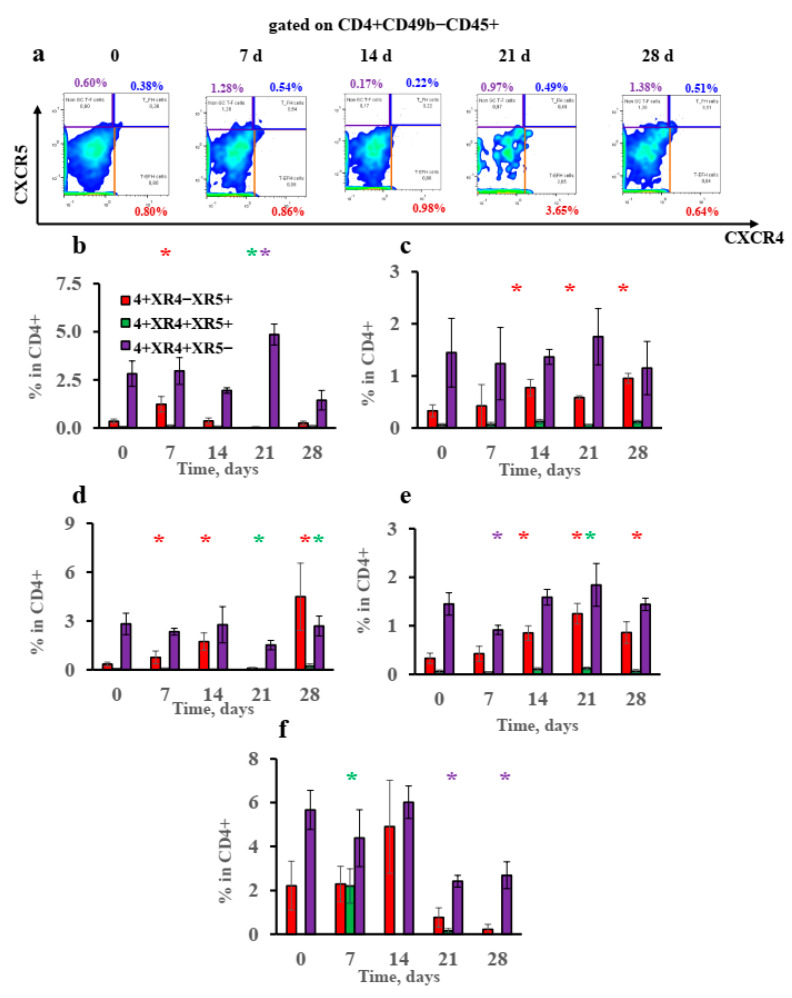
Unique pattern of humoral immune response and early induction of IgE B-cell class switching in withers subcutaneous fat upon low-antigen-dose administration may be linked with accumulation of extrafollicular T helpers and the absence of the accumulation of T helpers associated with B-cell follicles and follicular T helpers. BALB/c mice (*n* = 4 for each group–time point) were immunized by the SC route in the withers region (**a**–**e**) or by the IP route (**f**) with low (100 ng) or high (10,000 ng) OVA doses for 4 weeks, 3 times a week. Representative flow cytometry pseudocolor plots of subcutaneous fat cells gated on CD4+ CD49b- CD45+ T helpers (**a**). Percentage of CD4+ CXCR4-CXCR5+ T helpers situated in B-cell follicles, CD4+ CXCR4+ CXCR5+ follicular T helpers, and CD4+ CXCR4+ CXCR5- extrafollicular T helpers at indicated time points are shown for low (**b**,**c**,**f**)- and high-antigen-dose (**d**,**e**)-immunized mice in withers subcutaneous fat (**b**,**d**), regional lymph nodes close to withers region (**c**,**e**), and abdominal fat (**f**). Asterisks indicate significant differences (*p* < 0.05) in the percentage of the B-cell subpopulation, as shown by the same color with the intact group. Each asterisk of a certain color corresponds to the significance of the difference of subpopulation percentage, which is shown by a column of the same color. All experiments were performed 3 times, and representative data are shown.

## Data Availability

The data presented in this study are available in figures within the article and in Appendix A.

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
