# Peer review of "Early IgE Production Is Linked with Extrafollicular B- and T-Cell Activation in Low-Dose Allergy Model"

_vaccines, 2022, doi:10.3390/vaccines10060969_

Round 1

Reviewer 1 Report

The presented manuscript entitled "The IgE production is initially induced in subcutaneous fat and depends on extrafollicular B cells" is an interesting study and it could be beneficial for the readers of the "Vaccines" journal. The article is well written and discussed. I recommend it for acceptance for publication.

Author Response

We are very grateful to the reviewer for such a high assessment of our work and for his recommendation of our manuscript to the publication.

Reviewer 2 Report

In the current study, the authors revisit prior published observations verifying an alternate murine in vivo model for allergy IgE production.  Alternate models that more accurately reflect the response of an allergic individual are much needed. 

A major limitation to the study is the lack of a significant advance over prior published findings.  The alternate model was established and published in reference 28 and findings repeated herein. 

Other concerns regarding the manuscript in its current form is as follows:

  1. All flow cytometry related data  require representative flow plots to be shown within the figures in the main manuscript.
  2. Figures and the text need to be checked for proper call outs. 
  3. The gating strategy for ILCs is unconventional.  Lineage negative alone is not sufficient to identify ILC2 cells.  Lineage + has incorrectly been designated ILC2+ .  The relevance to the current study is also unclear. 
  4. The gating strategies in the supplemental figures are difficult to next to impossible to follow and gate percentages should be on top of gates versus in a row above the flow plot. 

Author Response

Point 1. A major limitation to the study is the lack of a significant advance over prior published findings.  The alternate model was established and published in reference 28 and findings repeated herein.

We are totally agree with the reviewer that many patterns of the immune response to low and high allergen doses administrated by different routes discussed in the present paper were presented also in the previous study. However, it should be noted that in this work, the participation of various subpopulations of B- and T-cells in the immune response to the administered low and high doses of the allergen in different ways was considered. Moreover we have done the kinetic analysis of their accumulation in different sites as well as the kinetic analysis of transcripts and excision circles’ linked with IgE isotype switching and IgE production. These kinetic analyses as well as the analysis of accumulation of different B- and T-cell subpopulations were not present in our previous work. From our previous work it was not completely clear what is the reason for enhanced IgE production upon immunization of mice by subcutaneous route in the withers region compared with intraperitoneal route. Also it was not clear how the earlier induction of antibody class switching in subcutaneous fat tissue compared with regional lymph nodes could be linked with different immune cell subpopulations activated in these different sites. From our present work it becomes clear that the enhanced early B-cell IgE class switching and IgE production may be linked with both extrafollicular B- and T-cell accumulation in subcutaneous fat. This was the main conclusion from the present work. This is because of in regional lymph nodes in close proximity to withers region where IgE class witching is delayed low antigen doses induce accumulation only extrafollicular plasmablasts but not extrafollicular T-helpers. Also in abdominal fat tissue where IgE class switching was also delayed we could not observe stable extrafollicular B- and T-cell accumulation.

Point 2. All flow cytometry related data require representative flow plots to be shown within the figures in the main manuscript.

The representative flow cytometry plots were added from Figure A2 to the main figures (Figure 4 and Figure 6).

Point 3. Figures and the text need to be checked for proper call outs.

We checked the references to the figures in the text and corrected the mistakes.

Point 4. The gating strategy for ILCs is unconventional. Lineage negative alone is not sufficient to identify ILC2 cells. Lineage + has incorrectly been designated ILC2+ The relevance to the current study is also unclear.

We are sorry for the mistake on Figure A3 (currently this is Figure A2) about incorrectly designated Lineage+ cells. We totally agree with reviewer that Lineage negative is not sufficient to indentify ILC2 cells, but we determinate ILC2 as CD45+Lin-ST2+ cells and this was described in Methods. However we agree with the fact that in current allergy model these cells have no sufficient impact. So we deleted them from gating strategy.

Point 5. The gating strategies in the supplemental figures are difficult to next to impossible to follow and gate percentages should be on top of gates versus in a row above the flow plot.

We thank to the reviewer for this advice. We corrected Figure 4, Figure 6 and supplementary figures with gating strategy and shown the percentages of the important cell subpopulations on the top of the gates or in close proximity to cytometry plots in the cases when there was no enough space for them on the gates. We also changed all contour plots to pseudocolor ones and made a color designations of the most important gates and signatures of cell subpopulations. We hope that it will make our figures easier to understand.

Reviewer 3 Report

The manuscript by Chudakov et al, The IgE production is initially induced in subcutaneous fat and depends on extrafollicular B cells" is an important study in the field that may help to understand the allergic response to persons that are predisposed to allergens. The authors have immunized the BALB/c mice  with low and high doses of model antigen OVA and measured the IgE production as well the investigated the pattern of B and T cell activation at different time points. The authors conclude that  in the subcutaneous adipose tissue the response to low antigen doses is extrafollicular by its nature and based primarily on the activation of CD19+B220-CD38-CD95+ plasmablasts and accumulation of CXCR4+CXCR5- extrafollicular  T- helpers is also critical for their formation. This unique type of immune response creates specific pattern of humoral pattern which is characterized by high IgE production accompanied by minimal IgG1 one and leads to allergy development However, I have following comments which can significantly improve the manuscript.

-I would suggest changing the title of the manuscript to make it clearer and more convincing.

-The abstract of the manuscript does not make sense and the authors should rewrite the abstract of the manuscript to make it clearer for the audience.

- The authors need to improve the quality of presentation of the manuscript especially the structure and flow of the manuscript to make it more understandable to the audience in an easy and clear language.

- Methodology needs to be explained in detail.

-line 294-295- there are typos and in rest of the manuscript. The authors should carefully read the manuscript before submission.

-The English language must be further improved for clarity.

Author Response

Point 1. I would suggest changing the title of the manuscript to make it clearer and more convincing.

We are grateful to the reviewer to his comments. We changed the title of the present manuscript to “Early IgE production is linked with extrafollicular B- and T-cell activation in low-dose allergy model” to emphasize the main idea of the work on the relationship of extrafollicular activation with IgE production in our alternative allergy model.

Point 2. The abstract of the manuscript does not make sense and the authors should rewrite the abstract of the manuscript to make it clearer for the audience.

We rewrote the abstract to emphasize the main idea of our manuscript more clearly. 

Point 3. The authors need to improve the quality of presentation of the manuscript especially the structure and flow of the manuscript to make it more understandable to the audience in an easy and clear language.

We changed some complex sentences in the Results and Discussion sections to more simple ones. We also have used MDPI Specialist English editing service to correct the style and flow of our manuscript. Due to the comment about Introduction part of our article made by reviewer in the table we added some extra references on the research works where the question about the participation of germinal centers and extrafollicular cells in IgE production was considered. We have also added a small part in which we indicate the purpose and explain the main logic of the experiments of this work.

Point 4. Methodology needs to be explained in detail.

We added a more detailed explanation and description of methods used in this work, namely: ELISA, RNA extraction and purification method, histology.

Point 5. line 294-295- there are typos and in rest of the manuscript. The authors should carefully read the manuscript before submission.

We corrected these typos in the manuscript. We are sorry for such mistakes.

Point 6. The English language must be further improved for clarity.

We are sorry for our poor knowledge of English. We used a Specialist English editing service in MDPI to improve our English after receiving this comments.

Round 2

Reviewer 2 Report

Authors have addressed my prior concerns.  Recommend to trim the introduction as it is lengthy.  Please also proof all facs plots for proper axis labels (one of the supplemental has live dead as FSC or SSC). 

Author Response

Point 1. Recommend to trim the introduction, as it is lengthy.

Response 1. We thanks to the reviewer for this advice. We shortened the "Introduction" section and deleted some accessory references.

Point 2. Please also proof all facs plots for proper axis labels (one of the supplemental has live dead as FSC or SSC).

Response 2. We are sorry for this typo, we corrected it by replacing «SSC-A» to «DAPI» on the axis label.

Reviewer 3 Report

The authors have made changes that has significantly improved the manuscript.

Author Response

Point 1. The authors have made changes that has significantly improved the manuscript.

Response 1. We thanks to the reviewer for the advices about improving our style and English language. However, we used Specialist English editing service to improve our article. We upload the certificate within the supplementary files. We also rewrote and simplified some sentences in the Results section of the article.